# Diagnostic Properties of Three SARS-CoV-2 Antibody Tests

**DOI:** 10.3390/diagnostics11081441

**Published:** 2021-08-10

**Authors:** Suelen Basgalupp, Giovana dos Santos, Marina Bessel, Lara Garcia, Ana Carolina de Moura, Ana Carolina Rocha, Emerson Brito, Giovana de Miranda, Thayane Dornelles, William Dartora, Lucia Pellanda, Pedro Hallal, Eliana Wendland

**Affiliations:** 1Hospital Moinhos de Vento, Porto Alegre 90035-000, RS, Brazil; suelen.basgalupp@hmv.org.br (S.B.); giovana.tavares@hmv.org.br (G.d.S.); marina.bessel@hmv.org.br (M.B.); lara.garcia@hmv.org.br (L.G.); ana.rocha@hmv.org.br (A.C.R.); emerson.brito@hmv.org.br (E.B.); giovana.miranda@hmv.org.br (G.d.M.); thayane.dornelles@hmv.org.br (T.D.); william.dartora@hmv.org.br (W.D.); 2Department of Public Health, Universidade Federal de Ciências da Saúde, de Porto Alegre, Porto Alegre 90050-170, RS, Brazil; anacarol.demoura@gmail.com (A.C.d.M.); lupellanda@gmail.com (L.P.); 3Postgraduate Programme in Epidemiology, Universidade Federal de Pelotas, Pelotas 96020-220, RS, Brazil; prchallal@gmail.com

**Keywords:** SARS-CoV-2, COVID-19, serological tests, antibodies, point-of-care test, ELISA, LUMIT

## Abstract

Serological assays emerged as complementary tools to RT-PCR in the diagnosis of SARS-CoV-2 as well as being needed for epidemiological studies. This study aimed to assess the performance of a rapid test (RT) compared to that of serological tests using finger prick blood samples. A total of 183 samples were evaluated, 88 of which were collected from individuals with negative RT-PCR and 95 from positive RT-PCR individuals. The diagnostic performance of RT (WONDFO^®^) and LUMIT (PROMEGA^®^) were compared to that of ELISA (EUROIMMUN^®^) for detecting antibodies against SARS-CoV-2 according to time from symptoms onset. The IgG antibody tests were detected in 77.4% (LUMIT), 77.9% (RT), and 80.0% (ELISA) of individuals. The detection of antibodies against SARS-CoV-2 increases in accordance with increasing time from symptoms onset. Considering only time from symptoms onset >21 days, the positivity rate ranged from 81.8 to 97.0% between the three tests. The RT and LUMIT showed high agreement with ELISA (agreement = 91.5%, k = 0.83, and agreement = 96.3%, k = 0.9, respectively) in individuals who had symptoms 15 to 21 days before sample collection. Compared to that of the ELISA assay, our results show sensitivity ranged from 95% to 100% for IgG antibody detection in individuals with symptoms onset between 15 and 21 days before sample collection. The specificity was 100% in individuals with symptoms onset >15 days before serological tests. This study shows good performance and high level of agreement of three immunoassays for the detection of SARS-CoV-2 antibodies.

## 1. Introduction

The new coronavirus disease 2019 (COVID-19) caused by severe acute respiratory syndrome coronavirus 2 (SARS-CoV-2) emerged in 2019 and quickly spread, causing a worldwide pandemic [1]. To date, 170,427,567 people were infected by SARS-CoV-2 and 3,543,311 deaths were recorded [2]. The rapid advance and dimension of the disease brought the necessity to adopt fast and effective measures to contain the virus through clinical and epidemiological actions, based mainly on the diagnostic capacity [3]. 

Patients with COVID-19 present a wide range of symptoms, ranging from asymptomatic to severe illness. Signs and symptoms may appear 2 to 14 days after exposure to the virus and the common clinical signs can include: fever, cough, shortness of breath or difficulty breathing, fatigue, muscle or body aches, headache, loss of taste or smell, sore throat, congestion or runny nose, nausea or vomiting, and diarrhea [4]. 

Two types of COVID-19 test are available: those detecting SARS-CoV-2 (viral particles/ active infection), such as reverse transcription polymerase chain reaction (RT-PCR), and those detecting the immune response of the body to infection (past exposure to the virus) called serological tests. The gold standard for the diagnosis of COVID-19 (identifying patients with acute SARS-CoV-2 infection as well as asymptomatic carriers) is the RT-PCR from a nasopharyngeal or oropharyngeal swab or bronchoalveolar lavage specimens [5]. 

Given the growing COVID-19 pandemic, serological tests are needed for epidemiological studies. These tests were developed to detect specific antibodies—IgA, IgM and IgG—against SARS-CoV-2 virus in human whole blood, serum, or plasma samples. Two main kinds of serological tests are available: enzyme-linked immunosorbent assay (ELISA) and lateral flow immunochromatographic assays (LFIAs) called point-of-care (POC) tests [6]. Numerous LFIAs were introduced into the market since SARS-CoV-2 emerged and were used as an alternative to nucleic acid amplification tests (NAATs) to assess infection [7,8,9,10]. 

In the context of population testing, there are important issues that need to be evaluated both from insufficient diagnostic sensitivity (leading to missing infected individuals) or insufficient diagnostic specificity (imposing confinement measures on individuals who are not infected). 

The aim of this study was to assess the performance of a rapid test (RT) compared to that of ELISA and LUMIT serological tests using finger prick blood samples in participants with SARS-CoV-2 RT-PCR test.

## 2. Materials and Methods

### 2.1. Study Population

This is a cross-sectional observational study that evaluated individuals with and without COVID-19 infection detected by RT-PCR who were invited through public media for a serological test in a Drive-thru model. This study was carried out during April–May 2020. 

The samples of the LFIA test were made using two drops of whole blood from the finger prick, and an aliquot of blood was stored in a microtainer^®^ tube (Becton, Dickinson and Company, Franklin Lakes, NJ, USA). After collection, in the laboratory this tube was centrifuged to obtain the serum. 

Participants answered a short questionnaire with sociodemographic information (sex, age, education, ethnicity, among others), symptoms and presence of comorbidities. The date of symptoms onset in these data refers to the date reported by the patient on which the clinical symptoms first appeared, where the clinical symptoms include fever, sore throat, cough, cougar in cough, difficulty breathing, stuffy nose, vomiting, and diarrhea. This study was approved by the National Committee of Ethics in Research (CONEP) (protocol code 30415520.2.0000.5313, 7 April 2020) and written informed consent was obtained from all participants.

### 2.2. Enzyme-Linked Immunosorbent Assay (ELISA) 

Serum samples were analyzed by the Euroimmun Anti-SARS-CoV-2 ELISA kit (Euroimmun Medizinische Labordiagnostika, Lübeck, Germany; Cat # EI 2668-9601 A and EI 2606-9601 G respectively), which detects IgA and IgG antibodies using recombinant S1 domain of the SARS-CoV-2 spike protein and interpreted according to the manufacturer’s instructions. Briefly, 1:101 diluted serum samples were added to wells coated with SARS-CoV-2 antigens and incubated at 37 °C for 60 min. After incubation, the wells were washed three times. In the case of positive samples, specific antibodies bound to the antigens. To detect the antigen-antibody bonds, a second incubation at 37 °C for 30 min was carried out using an enzyme conjugate, which was labelled anti-human IgA or IgG, and catalyzing a color reaction. Later, the excess of conjugate and the unreacted antibody were removed from the wells using a wash buffer. Then each well received a substrate solution and was incubated for 30 min at room temperature protected from direct sunlight. After the last incubation, the stop solution was added into each of the microplate wells. The amount of this bound to the antibody determined the color intensity, which in turn was measured by absorbance at 450 nm using an ELISA microplate reader (SpectraMax^®^ M3, Molecular Devices LLC, San Jose, CA, USA). 

The ratio between sample absorbance and calibrator on each plate was calculated and the results were evaluated semiquantitatively. According to the manufacturer’s recommendations, a ratio <0.8 is considered negative, ≥0.8 to <1.1 borderline, and ≥1.1 positive [11,12].

### 2.3. Rapid Lateral Flow Test

The presence of antibodies against SARS-CoV-2 was assessed using a lateral flow point-of-care test, which detects a single line qualitative IgG and IgM, but without distinction between them, the RT WONDFO^®^ SARS-CoV-2 Antibody Test (Wondfo Biotech, Guangzhou, China), using capillary whole blood samples. This lateral flow test detects IgM and IgG isotypes that are specific to the SARS-CoV-2 receptor binding domain of spike protein and this assay does not discriminate between IgM and IgG. The RT was performed by trained nurses, according to the manufacturer’s instructions, collecting two drops of blood from a finger prick after discarding the first drop. The test was read from 15 to 20 min after the addition of diluent. All tests with visible bands in test (T), and control (C) cassette were considered positive. If the C line does not appear, the test is invalid and should be repeated with a new cassette [13]. Additionally, 600 µL of capillary blood were collected in a BD Microtainer^®^ Gel Tube, centrifuged and serum was stored at −80 °C. The specificity and sensitivity of the RT compared to that of ELISA were calculated.

### 2.4. Lumit™ Dx SARS-CoV-2 Immunoassay

Serum samples were analyzed by the Lumit™ Dx SARS-CoV-2 Immunoassay (Promega Corporation, Madison, WI, USA, Cat.# VB1080). This immunoassay is based on NanoLuc^®^ Binary Technology (NanoBiT^®^), which is a luminescent structural complementation system designed for biomolecular interaction studies. It is composed of two subunits, Large BiT (LgBiT; 18 kDa) and Small BiT (SmBiT; 11 amino acid peptide), that were optimized for stability and minimal self-association due to weak affinity (190 μM). In the Lumit™ Dx SARS-CoV-2 Immunoassay, labeled SARS-CoV-2 protein is supplied in two forms; one form is labeled with SmBiT subunit, and the other with LgBiT subunit. In the presence of SARS-CoV-2 antibodies, the CoV-2-SmBiT and CoV-2-LgBiT proteins bind to the two Fab domains of the antibody, bringing LgBiT and SmBiT subunits into proximity. The two subunits then reassemble into a functional luminescent enzyme and generate a luminescent signal in the presence of Lumit™-Dx Detection Reagent. 

This immunoassay is designed for qualitative measurement of SARS-CoV-2 antibodies in human serum. It utilizes Lumit™ technology, a solution-based, high-throughput immunoassay with no washing steps. Briefly, CoV-2-SmBiT and CoV-2-LgBiT were added to a 96-well white plate, diluted serum samples, positive control, negative control, and calibrator were added to the wells, and the plate was incubated for 30 min at room temperature. During the incubation, the CoV-2 LgBiT and CoV-2 SmBiT bound to SARS CoV-2 antibodies and were brought into proximity, resulting in complementation and formation of a functional luminescent enzyme. Lumit™-Dx Detection Reagent was then added, and the luminescent signal was measured using a luminometer (GloMax^®^ Navigator, Promega). The luminescent signal was proportional to the SARS-CoV-2 antibodies in the serum sample. The results were calculated according to the ratio of the relative light unit (RLU) signal of the test sample (S) to the mean RLU signal of the Calibrator (C). If S/C ≥ 1, then the sample is positive for SARS-CoV-2 antibodies. If S/C < 1, then the sample is negative for SARS-CoV-2 antibodies [14].

### 2.5. Statistical Analysis

Continuous variables were analyzed using means and standard deviation, while categorical variables were expressed as absolute frequencies and percentages. Comparisons between variables were made using Chi-squared test. For these comparisons, a *p*-value less than 0.05 was considered significant. 95% confidence intervals (CIs) were calculated for agreement, sensitivity, specificity, predictive positive value (PPV), and predictive negative value (PNV). 

To assess the analytical properties of RT (Wondfo) the results were compared to ELISA assay, which is the gold standard for serological tests. Samples were stratified in three categories according to the time from symptoms onset: <15 days, 15–21 days, and >21 days.

Agreement between different serological tests was evaluated using Cohen’s kappa score. Cohen’s kappa (k) was classified as follows: 0.00–0.20, slight; 0.21–0.40, fair; 0.41–0.60, moderate; 0.61–0.80, substantial; 0.81–1.00, almost perfect [15]. Statistical analysis was carried out using SAS software (Statistical Analysis System, SAS Institute Inc., Cary, NC, USA), version 9.4 and R software, version 4.0.3.

## 3. Results

Over the study period, 183 individuals completed the survey and consented to participate in the serological test for COVID-19. The age ranged from 19 to 89 years (mean 47.7 (±14.08) years), and 104 (56.83%) were female. These 183 samples from RT-PCR positive and negative participants were collected by 0 to 47 days after RT-PCR testing, and all performed the Wondfo test and the blood collection. The sociodemographic and clinical characteristics of individuals who provided blood samples are shown in Table 1. Fever was the only symptom that showed a statistically significant difference between the RT-PCR positive and RT-PCR negative groups (*p* < 0.01).

Regarding symptoms, of the 183 participants evaluated, 138 reported having at least one symptom of COVID-19. Thus, the participants were divided into 3 categories: 36 individuals (26.09%) had symptoms up to 15 days before sample collection, 47 individuals (34.06%) had symptoms between 15 and 21 days before, and 55 individuals (39.85%) had symptoms after 21 days before sample collection for serological testing.

Of the total participants, 88 (48.1%) of which were collected from individuals who had tested negative SARS-CoV-2 by RT-PCR and 95 (51.9%) from individuals with a positive SARS-CoV-2 RT-PCR result in respiratory specimens. From negative SARS-CoV-2 RT-PCR, 83 tested negative using the RT (IgM and/or IgG) (94.32% agreement, 95% CI: 89.43–99.20), and from positive SARS-CoV-2 RT-PCR participants, 74 tested positive for IgM and/or IgG (77.89% agreement, 95% CI: 69.47–86.32).

Regarding the ELISA (Euroimmun), of these 95 patients tested positive for RT-PCR, 76 tested positive (80.00% agreement, 95% CI: 71.88–88.12). Of these 88 tested negative for RT-PCR, and 84 tested negative for IgG (95.45% agreement, 95% CI: 91.06–99.85) (as illustrated in Figure 1). The proportion of samples testing positive for ELISA test and distribution of antibodies according to time since onset of symptoms is demonstrated in Figure 2. 

Overall, the IgG antibody tests were detected in 77.4–80.0% of individuals. The detection of antibodies against SARS-CoV-2 shows a growth in accordance with the increasing time from symptoms onset. When the time from symptoms onset >21 days was taken into consideration, the positivity rate was 97.0%, 90.6% and 81.8% for ELISA, LUMIT and RT, respectively (as illustrated in Table 2). 

The proportion of RT-PCR positive individuals with antibodies against SARS-CoV-2 detected by ELISA (IgG), LUMIT, and RT according to two symptoms onset categories (≤21 and >21 days) is demonstrated in Figure 3. Our results show that for each test evaluated, the proportion (in percent) of RT-PCR positive individuals who had SARS-CoV-2 antibodies detected was higher considering symptoms onset >21 days compared to up to 21 days of symptoms onset (as illustrated in Figure 3).

The diagnostic performance of three immunoassays for detecting antibodies against SARS-CoV-2 was evaluated according to time from symptoms onset. RT (Wondfo) and LUMIT (Promega) were compared to ELISA (Euroimmun), which is considered the gold standard for serological tests.

For RT and LUMIT, the sensitivity was 100% when the time from symptoms onset was 15 to 21 days. However, the specificity of RT and LUMIT was 100% considering >21 days between the onset of symptoms and the serological test. When we compare the RT with LUMIT, the sensitivity was 100% considering the period from 15 to 21 days from symptoms onset. When the time from symptoms onset >21 days was taken into consideration, the specificity was 100% (as illustrated in Table 3). 

The diagnostic performance of three immunoassays for detecting antibodies against SARS-CoV-2 was also evaluated according to RT-PCR positive individuals. RT (Wondfo) and LUMIT (Promega) were compared to ELISA (Euroimmun), which is considered the gold standard for serological tests.

Compared to ELISA, LUMIT and RT tests showed the same sensitivity (88%) in RT-PCR positive individuals. LUMIT showed a high specificity (100%) compared to that of RT (63%). The sensitivity of the RT compared to that of the LUMIT was slightly higher than when compared to that of the ELISA, but the specificity was reduced (as illustrated in Table 4).

To determine the agreement between IgG serological assays evaluated, Cohen’s kappa score was calculated according to the time from onset of symptoms. The RT and LUMIT showed high agreement with ELISA (agreement = 91.5%, k = 0.83, and agreement = 96.3%, k= 0.9, respectively) in individuals who had symptoms 15 to 21 days before sample collection. 

## 4. Discussion

This is the first study to evaluate the performance of the LUMIT assay. Compared to that of ELISA, our results show high sensitivity for IgG antibody detection in individuals with symptoms onset 15 to 21 days before sample collection and good specificity in individuals with symptoms onset >21 days before serological tests. This method is easy to execute compared to that of the ELISA and the total assay time is less than 1 h. It is a high-throughput immunoassay with no washing steps and requires only a luminescent microplate reader for signal detection. Using ELISA as standard, our results show that LUMIT detects only IgG antibodies.

In this context of scarcity of vaccines for massive immunization and limited treatment options for COVID-19 worldwide, the validation of rapid serologic testing is required. Numerous rapid tests options have appeared on the market, however, due to the highly variable sensitivity and specificity of these assays for COVID-19 immunity, internal validation became necessary. Serologic assessment provides valuable information on past exposure using venous and capillary blood samples, although the protective effect of anti-SARS-CoV-2 antibodies remains uncertain [16].

We observe false positives and negatives, where false negative results on the RT-PCR for SARS-CoV-2 can occur due to problems in the preanalytical phase, collection practice, and collection at the time of symptoms, in addition to the RNA instability itself [17]. RT, ELISA and LUMIT presented 5, 4, and 2 false positive results, respectively. Two of them were common to all three tests, but the rest was test-specific. The number of false negative results for RT, ELISA, and LUMIT was 21, 19, and 19, respectively. Five of them were common to all three tests.

Furthermore, in our results, we observed a high number of negative cases in serological methods after positive RT-PCR, but these results can be expected, taking into account the symptoms presented as well as that identified by Yongchen et al., in which there was an immediate response in the seroconversion of critically ill patients and among asymptomatic patients, and only 20% presented seroconversion [18].

Our results showed, in the cut-off point of 21 days of symptoms, a higher percentage of IgG positive results in all methods used, representing the best time interval for seroconversion identification. Related to this percentage, we observed in a review study [19] that evaluated the following methods: immunochromatographic assay (RT), immunoenzymatic assay (ELISA), chemiluminescent assay (CLIA), and dry fluorescence. This review showed the best cutoff point, reaching 98.9% of seroconversion over 28 days. However, when they evaluated in 21 days, they identified 93% of seroconversion, while the ELISA (Euroimmun) technique showed 97% in the same period.

In addition, Traugott M. et al. found that in 11 days the seroconversion for both ELISA and the RT (Wantai SARS-CoV-2 Ab Rapid Test) by 100%, but in the same period the “2019-nCOV Rapid Test IgM” test showed 45.45 % while 2019-nCOV RapidTest: IgG also showed 100% seroconversion. Thus, we observed a great variation in the percentage of positivity presented between the different brands, and for the RT, Wondfo (WONDFO^®^ SARS-CoV-2 Antibody Test) showed 81% after 21 days [20].

Regarding the Lumit™ DX SARS-CoV-2 test, the kit’s manufacturing instructions show that after more than 20 days of symptoms, the test has a sensitivity of 93.5% of the Lumit™ Dx SARS-CoV-2 in relation to the RT-PCR, our results showed a high sensitivity of Lumit™ (97%), but low specificity (50%), which can be explained once the samples were frozen for about 4 months until the test, with two defrosts. 

The sensitivity for the detection of IgG antibodies after 14 days from onset of symptoms was > 92% for seven different rapid tests, compared to 89.5% for the IgG ELISA (Euroimmun) [21]. Different assays were also compared: automated ELISA (Euroimmun) test or chemiluminescence enzyme immunoassays or rapid detection test (lateral flow immunoassays); the sensitivity after 14 days of symptoms was 100% for all of them [22]. These results corroborate ours; when we compare all the three immunoassays after 21 days from symptoms onset, there is 100% sensitivity. Also, comparing ELISA and LUMIT, the specificity was 100% for all the analyzed periods. When we compare the RT to ELISA or LUMIT, the sensitivity was 100% considering the period from 15 to 21 days from symptoms onset, and an overall sensitivity >88%.

Being a complement to RT-PCR, the detection of antibodies can give additional information to the diagnosis of COVID-19. In the plasma samples of patients, from days 8 to 14 after the symptoms onset, ELISA tests for total antibodies (Ab), IgM, and IgG, showed that their sensitivities were all higher than that of RT-PCR. After 15 days from onset of symptoms, the sensitivities of Ab, IgM, and IgG were 100.0%, 94.3%, and 79.8%, respectively, contrasting with RNA that was only detectable in 45.5% [23]. Thus, it would be valid to compare the three immunoassays used in this study among them (see data in Table 3). As the seroconversion typically occurs 7–14 days after the onset of symptoms [24], and the samples we used for the different serological tests were all collected from the patients in the same day, we can relate the onset of symptoms to seroconversion comparing the three assays here analyzed. 

Our study had some limitations. First, not all the samples were available for the three tests. Case numbers in the tables may have small discrepancies; however, results were not compromised. Since the LUMIT assay utilizes a SARS-CoV-2 protein as bait, the procedure is not specific to any particular Ig class. However, the LUMIT assay did not present positive results to those samples that were positive in the IgA ELISA test. The LUMIT assay kits were donated by the Promega company, and we did not have sufficient LUMIT reagents to analyze all samples; therefore, just a subsample was analyzed. Another important aspect to mention is that our study did not evaluate cross-reactivity with other coronaviruses, which could generate false positive results in the serological determinations. Serological studies showed cross-reactivity of SARS-CoV-2 S protein with SARS-CoV (the agent responsible for the 2003 epidemic), MERS (Middle East Respiratory Syndrome coronavirus), and sCOVs (seasonal coronaviruses) [25,26,27,28,29,30].

## 5. Conclusions

Although molecular tests are the gold standard and very specific in early COVID-19 detection, the use of serological assays can provide the information about the immunological response by production of antibodies against SARS-CoV-2, according to the stage of the disease. The RT analysis showed consistency when compared to that of ELISA. These results demonstrate that it could be an adequate method for detection of antibodies in individuals who had SARS-CoV-2 infection at least 2 weeks before. This study reveals the good performance and the high level of agreement of three immunoassays for the detection of SARS-CoV-2 antibodies. The serological assays may be convenient to assess the immune response to vaccines and for development of seroepidemiological studies.

## Figures and Tables

**Figure 1 diagnostics-11-01441-f001:**
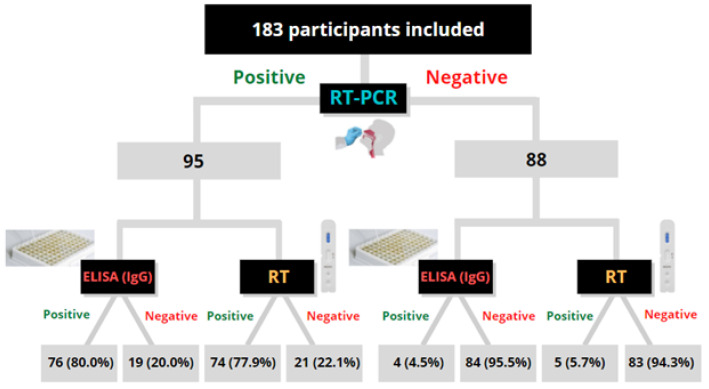
Schematic representation of results obtained from RT-PCR, RT, and ELISA tests.

**Figure 2 diagnostics-11-01441-f002:**
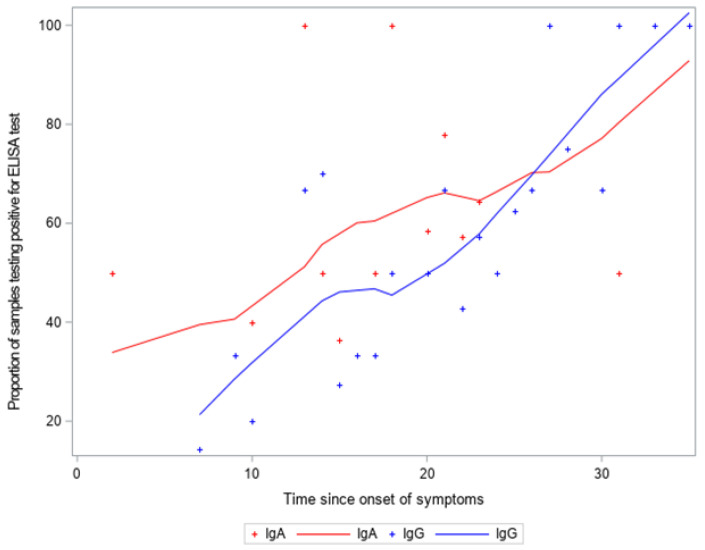
Proportion of samples testing positive for ELISA (Euroimmun) test: distribution of IgA + and IgG + according to time since onset of symptoms.

**Figure 3 diagnostics-11-01441-f003:**
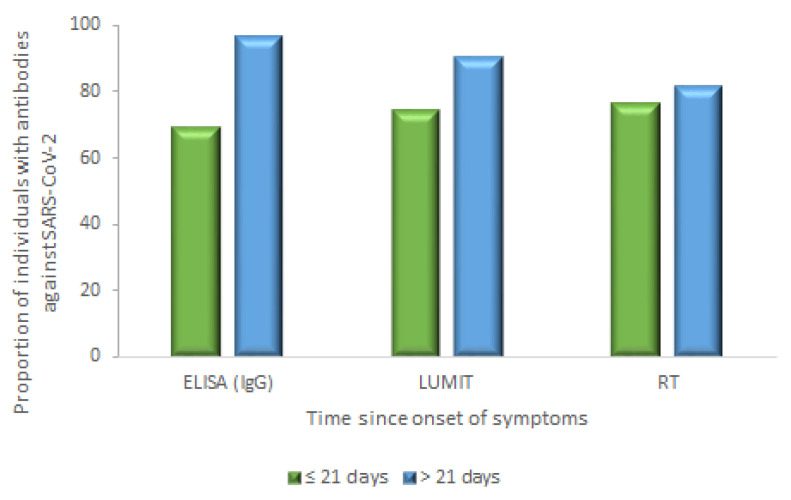
Proportion of RT-PCR positive individuals with antibodies against SARS-CoV-2 detected by ELISA (IgG), LUMIT, and RT according to two symptoms onset categories.

**Table 1 diagnostics-11-01441-t001:** Characteristics of included participants.

Characteristics	RT-PCR Positive N (%)	RT-PCR Negative N (%)	*p*-Value **
**Sex**			*p* < 0.01
Male	50 (52.6%)	29 (33.0%)	
Female	45 (47.4%)	59 (67.0%)	
**Age (years)**			*p* < 0.05 *
20–35	21 (22.1%)	20 (22.7%)	
36–45	19 (20.0%)	34 (38.6%)	
46–59	25 (26.3%)	19 (21.6%)	
60+	30 (31.6%)	15 (17.1%)	
**Ethnicity**			*p* = 0.01 *
White	91 (96.8%)	81 (92.1%)	
Black	0 (0.0%)	1 (1.1%)	
Brown	0 (0.0%)	6 (6.8%)	
Other	3 (3.2%)	0 (0.0%)	
**Education level**			*p* = 0.34
Elementary school	0 (0.0%)	1 (1.1%)	
Secondary school	5 (5.3%)	2 (2.3%)	
Graduate	90 (94.7%)	85 (96.6%)	
**Symptoms**			
Fever	48 (50.5%)	21 (24.1%)	*p* < 0.01 *
Sore throat	36 (37.9%)	43 (50.0%)	*p* = 0.13
Cough	48 (51.6%)	38 (44.2%)	*p* = 0.37
Cougar in cough	13 (27.1%)	8 (23.5%)	*p* = 0.80
Difficulty breathing	30 (31.6%)	24 (27.6%)	*p* = 0.63
Stuffy nose	14 (41.2%)	14 (58.3%)	*p* = 0.29
Vomiting	6 (6.5%)	9 (10.5%)	*p* = 0.42
Diarrhea	39 (41.1%)	33 (37.5%)	*p* = 0.65
**Comorbidities**			
Diabetes	5 (5.3%)	2 (2.3%)	*p* = 0.45
Asthma	9 (9.6%)	16 (18.4%)	*p* = 0.13
Hypertension	12 (12.6%)	9 (10.5%)	*p* = 0.82

* A *p* value less than or equal to 0.05 was considered significant. ** Chi-square-test.

**Table 2 diagnostics-11-01441-t002:** Proportion of individuals with antibodies against SARS-CoV-2 according to RT-PCR test, from onset of COVID-19 symptoms.

Test	Overall	Time from Symptoms Onset		
		<15 Days	15 to 21 Days	>21 Days
**ELISA (Euroimmun)**	(*n* = 138)	(*n* = 36)	(*n* = 47)	(*n* = 55)
IgA or IgG	89.5%	92.9%	75.9%	97.0%
IgA	85.3%	78.6%	75.9%	94.0%
IgG	80.0%	85.7%	62.1%	97.0%
**LUMIT (Promega)**	(*n* = 80)	(*n* = 17)	(*n* = 27)	(*n* = 36)
IgG	77.4%	76.9%	73.9%	90.6%
**RT (Wondfo)**	(*n* = 138)	(*n* = 36)	(*n* = 47)	(*n* = 55)
IgM or IgG	77.9%	78.6%	75.9%	81.8%

**Table 3 diagnostics-11-01441-t003:** Diagnostic performance of three immunoassays for detecting antibodies against SARS-CoV-2.

	Overall % (95% CI)	Time from Symptoms Onset		
		<15 Days % (95% CI)	15 to 21 Days % (95% CI)	>21 Days % (95% CI)
**RT vs. ELISA**	(*n* = 138)	(*n* = 36)	(*n* = 47)	(*n* = 55)
Sensitivity	88.0 (78.0–94.0)	83.0 (52.0–98.0)	100.0 (82.0–100.0)	85.0 (68.0–95.0)
Specificity	91.0 (84.0–96.0)	92.0 (73.0–99.0)	86.0 (67.0–96.0)	100.0 (85.0–100.0)
Positive predictive value (PPV)	89.0 (79.0–95.0)	83.0 (52.0–98.0)	83.0 (61.0–95.0)	100.0 (85.0–100.0)
Negative predictive value (NPV)	90.0 (83.0–95.0)	92.0 (73.0–99.0)	100.0 (86.0–100.0)	81.0 (62.0–94.0)
**LUMIT vs. ELISA**	(*n* = 80)	(*n* = 17)	(*n* = 27)	(*n* = 36)
Sensitivity	86.0 (76.0-93.0)	83.0 (52.0–98.0)	95.0 (74.0–100.0)	94.0 (79.0–99.0)
Specificity	100.0 (85.0-100.0)	100.0 (48.0–100.0)	100.0 (63.0–100.0)	100.0 (40.0–100.0)
Positive predictive value (PPV)	100.0 (85.0-100.0)	100.0 (69.0–100.0)	100.0 (81.0–100.0)	100.0 (88.0–100.0)
Negative predictive value (NPV)	67.0 (48.0-82.0)	71.0 (29.0–96.0)	89.0 (52.0–100.0)	67.0 (22.0–96.0)
**RT vs. LUMIT**	(*n* = 80)	(*n* = 17)	(*n* = 27)	(*n* = 36)
Sensitivity	93.0 (83.0–98.0)	90.0 (55.0–100)	100.0 (81.0–100.0)	90.0 (73.0–98.0)
Specificity	64.0 (45.0–80.0)	57.0 (18.0–91.0)	56.0 (21.0–86.0)	100.0 (54.0–100.0)
Positive predictive value (PPV)	84.0 (73.0–91.0)	75.0 (43.0–95.0)	82.0 (60.0–95.0)	100.0 (87.0–100.0)
Negative predictive value (NPV)	81.0 (61.0–93.0)	80.0 (28.0–99.0)	100.0 (48.0–100.0)	67.0 (30.0–93.0)

CI, confidence interval.

**Table 4 diagnostics-11-01441-t004:** Diagnostic performance of three immunoassays for detecting antibodies against SARS-CoV-2 according to RT-PCR positive individuals.

	RT vs. ELISA % (95% CI) (*n* = 95)	LUMIT vs. ELISA % (95% CI) (*n* = 84)	RT vs. LUMIT % (95% CI) (*n* = 84)
Sensitivity	88.0 (79.0–94.0)	88.0 (78.0–94.0)	92.0 (83.0–97.0)
Specificity	63.0 (38.0–84.0)	100.0 (69.0–100.0)	47.0 (24.0–71.0)
Positive predictive value (PPV)	91.0 (81.0–96.0)	100.0 (94.0–100.0)	86.0 (75.0–93.0)
Negative predictive value (NPV)	57.0 (34.0–78.0)	53.0 (29.0–76.0)	64.0 (35.0–87.0)

CI, confidence interval.

## Data Availability

Data are available by the corresponding author.

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
