# Peer review of "Diagnostic Properties of Three SARS-CoV-2 Antibody Tests"

_diagnostics, 2021, doi:10.3390/diagnostics11081441_

Round 1

Reviewer 1 Report

In this manuscript, the authors compared 3 different assays to measure serology based on whole blood fingertip samples. The manuscript was written clearly, but there are some critical considerations that should be addressed:

  1. It is important to note that the ELISA, RT and LUMIT are measuring different subclasses of immunoglobulins (Ig). Critically, the authors did not provide data to show what classes of Ig does LUMIT detect, although the authors hinted that it is IgG. If they are detecting different subclasses, perhaps a direct comparison between ELISA, RT and LUMIT is incorrect. Instead, it could be more appropriate to analyse the RT-PCR positive samples to reflect on the specificity and sensitivity, as in these patients, they must have been exposed to the virus before and thus, should produce antibodies.
  2. The authors provided results as positive and negative readouts, but did not present the data the magnitude of the readings, and how the "cut-off" was decided for the three different assays. Also, does the magnitude differences captured in both 3 assays??
  3. The ROC curves will need to be presented if the authors want to claim sensitivity and specificity of the assays.
  4. The LUMIT samples only comprised of 80 while the others are 138. This can influence the outcome of figure 3, because the subset of the sera chosen could coincidentally have overall more IgG, which will improve the sensitivity of assay. As such, Figure 3 interpretation is incorrect if the investigators did not measure the remaining 58 samples.

Reviewer 2 Report

In this manuscript, Basgallupp et al determined  the performance of a rapid test  and LUMIT in  detecting antibodies against SARS-CoV-2 compared to a standard ELISA in relation to time from symptoms onset.

The results indicate that detection of antibodies against SARS-CoV-2 increases with time from symptoms onset and, at  > 21 days from symptoms onset, the agreement between the different tests varies from 81.8 to 97.0%.

In general, the manuscript is very well written, the experimental design is clear, and the results are adequately discussed. However, there are some points that need to be addressed before publication of the manuscript.

1)Although the authors have RT-PCR data from patients, the whole study is designed on the time of symptoms onset, which in turn is based on a self-declaration of the patients and not on a formal clinical diagnosis. This makes the study highly unreproducible, because it is based on unverifiable data. In the presentation of the data in the tables and figures, the authors should therefore refer to the time elapsed from positivity to RT-PCR, which is certainly a more reliable data.

2) It would be useful to indicate and discuss whether the false positives and false negatives identified in the different types of tests correspond to the same patients or to different patients.

Round 2

Reviewer 1 Report

Based on the authors reply, I am still uncertain the detection range (ie limit of detection and saturation intensity). Can I kindly request the authors to present the standard curves for the different assays to allow readers to better appreciate the comparison between the 3 assays?

Author Response

Point 1: Based on the authors reply, I am still uncertain the detection range (ie limit of detection and saturation intensity). Can I kindly request the authors to present the standard curves for the different assays to allow readers to better appreciate the comparison between the 3 assays?

Response 1: Thank you for your comment. We did not do the standard curves for the different assays because they were not the purpose of the study. We used the detection limit of the tests established according to the manufacturer's instructions of each kit used in the study, which are in the following references: Anti-SARS-CoV-2 ELISA (IgA) protocol. EUROIMMUN®. Available online: https://testecovid19.org/wp-content/uploads/2018/10/Anti-SARS-CoV-2-ELISA-IgA.pdf?x45112 (accessed on 20 July 2021); Anti-SARS-CoV-2 ELISA (IgG) protocol. EUROIMMUN®. Available online: https://testecovid19.org/wp-content/uploads/2018/10/Anti-SARS-CoV-2-ELISA-IgG.pdf?x45112 (accessed on 20 July 2021); Wondfo SARS-CoV-2 Antibody Test (Lateral Flow Method) protocol. WONDFO® (Catalog N°.: W195). Available online:  https://www.zarenta.at/WNDF_EN_antikoerper_Rel20200528.pdf (accessed on 20 July 2021); Lumit™ Dx SARS-CoV-2 Immunoassay protocol. PROMEGA®. Available online: https://www.promega.com/-/media/files/resources/protocols/technical-manuals/500/lumit-dx-sars-cov-2-immunoassay-protocol-tm636.pdf?la=en (accessed on 20 July 2021). These references have been added in the methods section of the article.

Reviewer 2 Report

The authors responded convincingly to the criticisms raised; the manuscript is sound enough to be published.

Author Response

Point 1: The authors responded convincingly to the criticisms raised; the manuscript is sound enough to be published.

Response 1: Thank you for your comments.